# The shame spiral of addiction: Negative self-conscious emotion and substance use

**Abigail W. Batchelder**[1,2,3]*, **Tiffany R. Glynn**[4], **Judith T. Moskowitz**[5], **Torsten B. Neilands**[6], **Samantha Dilworth**[6], **Sara L. Rodriguez**[3], **Adam W. Carrico**[7]

**1** Department of Psychiatry, Harvard Medical School, Boston, Massachusetts, United States of America, **2** Department of Psychiatry, Massachusetts General Hospital, Boston, Massachusetts, United States of America, **3** The Fenway Institute, Fenway Health, Boston, Massachusetts, United States of America, **4** Department of Psychology, University of Miami, Coral Gables, Florida, United States of America, **5** Northwestern University Feinberg School of Medicine, Chicago, Illinois, United States of America, **6** Department of Medicine, Division of Prevention Science, Center for AIDS Prevention Studies (CAPS), University of California, San Francisco, San Francisco, California, United States of America, **7** University of Miami Miller School of Medicine, Miami, Florida, United States of America

* a.carrico@miami.edu

**Data Availability Statement:** All data used in this manuscript are included in the attached Supporting Information file titled 110 PLoS One.

## Abstract

### Background

The bidirectional associations between negative self-conscious emotions such as shame and guilt and substance use are poorly understood. Longitudinal research is needed to examine the causes, consequences, and moderators of negative self-conscious emotions in people who use substances.

### Methods

Using parallel process latent growth curve modeling, we assessed bidirectional associations between shame and guilt and substance use (i.e., number of days in the past 30 used stimulants, alcohol to intoxication, other substances, or injected drugs) as well as the moderating role of positive emotion. Emotions were assessed using the Differential Emotions Scale. The sample included 110 sexual minority cisgender men with biologically confirmed recent methamphetamine use, enrolled in a randomized controlled trial in San Francisco, CA. Participants self-reported emotions and recent substance use behaviors over six time points across 15 months.

### Results

Higher initial levels of shame were associated with slower decreases in stimulant use over time ($b = 0.23$, $p = .041$) and guilt was positively associated with stimulant use over time ($\beta = 0.85$, $p < .0001$). Initial levels of guilt and alcohol use were positively related ($b = 0.29$, $p = .040$), but over time, they had a negative relationship ($\beta = -0.99$, $p < .0001$). Additionally, higher initial levels of other drug use were associated with slower decreases in shame over time ($b = 0.02$, $p = .041$). All results were independent of depression, highlighting the specific role of self-conscious emotions.

**Funding:** This project was supported by the National Institute on Drug Abuse (R01-DA033854; Carrico, Woods, and Moskowitz, PIs) and the National Institute of Mental Health (K24-MH093225; Moskowitz, PI). Additional support for this project was provided by the University of California, San Francisco Center for AIDS Research's Virology Core (P30-AI027763; Volberding, PI) and the Center for HIV Research and Mental Health (P30-MH116867; Safren, PI). Dr. Batchelder's time was supported by the National Institute on Drug Abuse Award K23DA043418 (Batchelder, PI). The content of this work is solely the responsibility of the authors and does not necessarily represent the official views of the NIH. The NIH had no role in study design, data collection and analysis, decision to publish, or preparation of the article.

**Competing interests:** The authors have declared that no competing interests exist.

## Conclusions

Shame and guilt are barriers to reducing stimulant use, and expanded efforts are needed to mitigate the deleterious effects of these self-conscious emotions in recovery from a stimulant use disorder.

## Introduction

As substance use disorders remain a significant public health problem resulting in high global disability prevalence [1], increasing our understanding of their development and maintenance is critical. Informed by the negative reinforcement [2] and tension-reduction [3] models of addiction, negative emotions have long been hypothesized to play a central role in the etiology of substance use disorders. However, relatively little is known regarding the specific negative emotions that stem from problematic patterns of substance use and potentiate continued substance use. Although negative self-conscious emotions such as shame and guilt are prominent among those living with substance use disorders, the extent to which these social emotions function as a cause or consequence of substance use remains unclear [4].

While shame and guilt are often mentioned together, substantial research indicates that they are distinct emotions related to different attributions and behavioral responses [5]. Negative self-conscious emotions, including shame and guilt, result from appraisal of how an experience pertains to the self in relation to others [6]. Shame is often conceptualized as a negative evaluation of oneself (e.g., "I am less valuable than others") while guilt is often conceptualized as a negative evaluation of one's behavior (e.g., "I have behaved poorly compared to others") [7–9]. Behaviorally, shame has been associated with avoidance and, paradoxically, a desire to change one's self or one's behavior, as well as an alert to threats of social belongingness, while guilt has been associated with eliciting a reparative response, such as an apology [6, 10, 11]. While guilt is straightforward in terms of associated behaviors, the paradoxical response to shame is consistent with the divergent findings that higher shame is associated with both an avoidant or maladaptive response, such as higher likelihood of relapse among individuals recovering from alcohol use disorder [12], and a positive or adaptive response, such as a longer duration in substance use treatment [13]. Shame-free guilt has been correlated with characteristics such as enhanced empathy [14] and has been associated with increased likelihood of taking responsibility for one's actions rather than deflecting blame [15], leading some to conceptualize it as more adaptive than shame. Mechanistically, recent research indicates that guilt-proneness is associated with routine use of protective and harm-avoidant behaviors (e.g., limiting drinking, not exceeding a specific amount of alcohol, etc.) during episodes of alcohol intake, whereas shame-proneness is not related to these behaviors [16]. Interestingly, a recent meta-analysis of laboratory studies of shame indicated that the likelihood of shame eliciting an adaptive rather than a maladaptive response was dependent on individuals' perception of their ability to repair a positive sense of self [10]. Comparing shame to guilt in relation to substance use is of importance given that shame-proneness, or the tendency to feel bad about oneself, has been associated with substance use problems; whereas, guilt-proneness, or the tendency to feel bad about a specific behavior, has inconsistently been associated with substance use problems [17, 18]. More work is needed to assess how shame and guilt are similar and different in their relationships to substance use.

The "shame addiction cycle" refers to a pattern of substance use to escape or avoid negative self-conscious emotions that paradoxically leads to increased shame related to the stigma of

being a person who uses substances [7, 19, 20]. The experience of self-conscious emotions indicates social evaluative threat from negative appraisals by others, which may not only perpetuate substance use but also elicit physiological stress, including triggering the hypothalamic-pituitary-adrenal axis stress responses, such as initiating inflammation [21]. Notably, pro-social aspects of negative self-conscious emotions have been conceptualized from an evolutionary perspective, including theorization that shame and guilt function to restrain human behavior [22] or maintain cultural standards [23]. From this perspective, shame and guilt may have the capacity to reduce the proposed cyclical relationship between negative self-conscious emotion and substance use.

Despite the potential benefits of negative self-conscious emotions, they have long been described in relation to alcohol use disorder (e.g., [24, 25]) and treatment (e.g., [26]). However, only two studies have explicitly investigated negative self-conscious emotions and substance use as antecedents and consequences of one another, both of which focused on shame and alcohol use. The first study identified that college students who described themselves as drinking more than their peers reported shame after drinking, which was then associated with increased drinking over the subsequent week [27]. The second study identified that higher levels of shame during the day were associated with higher likelihood of solitary drinking; however, level of drinking was only weakly related to shame the following day in a community sample [19]. While these two studies indicate that shame may be bidirectionally associated with alcohol use, no studies we are aware of have assessed bidirectional relationships between both shame and guilt and substance use beyond alcohol over time. Further, these studies involve college students or community samples, and the detrimental effects of self-conscious emotions may differentially affect substance use among populations with multiple, co-occurring stigmatized identities such as sexual minority men (i.e., gay, bisexual, and other men who have sex with men) living with HIV [28, 29].

As the empirical evidence for the cyclical relationships between negative self-conscious emotions and substance use is inconsistent, investigation of the bidirectional associations are needed. This is exemplified in the findings from a recent meta-analysis that found that shame was not significantly associated with substance consumption but was associated with substance use dependence and related symptoms (e.g., frequency of use; [4]). The authors venture that this may be because at times shame promotes and at other times it inhibits substance use, resulting in the average association yielding an observed effect near zero. Consistently, others have identified ways in which shame prevents or mitigates substance use (e.g., the stigma associated with substance use may act as a deterrent from initiation or perpetuation of substance use; [30]). Further, Luoma and others have noted that the negative consequences of problematic substance use also elicit shame [31]. Notably, the vast majority of research investigating self-conscious emotions and substance use focuses on shame in relation to alcohol use, limiting the ability to investigate different relationships between shame and guilt and types of substance use. Together, these inconsistent findings make the assessment of bidirectional longitudinal relationships between specific negative self-conscious emotions and types of substance use particularly important for understanding how shame, guilt, and substance use relate to one another.

In addition to relationships between substance use and negative self-conscious emotions, positive emotion has been identified as a potential moderator. The revised stress and coping theory indicates that positive and negative emotions may work together to influence individuals' behavior [32, 33]. Specifically, positive emotions following, or within the context of, negative emotions are thought to facilitate approach behavior [34–36], including engaging in one's environment and partaking in activities and may have unique, adaptive benefits for reducing stress reactivity [33]. Several studies have identified interactions between negative and positive

emotions in relation to substance use. For example, Mohr and colleagues [37] found that positive emotions buffered the relationship between shame and drinking among a sample of undergraduates who drink alcohol. Additionally, positive emotion was found to moderate the relationship between negative self-conscious emotion and HIV transmission risk behavior among a sample of individuals recently diagnosed with HIV [38], indicating that positive emotion may enable the acknowledgement and utilization of negative self-conscious emotions to reduce risk behaviors. Interactions between negative self-conscious and positive emotions may account for the divergent findings related to emotion and substance use.

While much of the research investigating the relationships between negative self-conscious emotions and substance use has focused on alcohol use among college students, research is needed to better understand differential relationships between shame and guilt and substance use among other populations. Recent evidence indicates that among individuals living with co-occurring stigmatized identities, such as sexual minority men living with HIV who use substances, substance use stigma, defined broadly to include stigma related to alcohol and other substance use, has been described as the most burdensome stigma, beyond stigma related to sexual orientation and HIV [39]. Additionally, substance use stigma, beyond stigma related to sexual orientation and HIV, has been identified as a barrier to engagement in healthcare among sexual minority men and people living with HIV who use substances (e.g., [40, 41]).

In the present longitudinal study, we investigated the bidirectional associations of negative self-conscious emotions with substance use, including stimulants, alcohol, other substances, and injection drug use among a sample of sexual minority men living with HIV. We examined shame and guilt separately given the literature indicating their unique psychological experiences [5], behavioral sequelae [6, 10, 11], as well as their differential associations with addiction-related behaviors [16–18]. Then, we determined whether positive emotions buffered, or moderated, these relationships. Investigating the causes, consequences, and moderators of shame and guilt separately in relation to substance use represents a critical step in clarifying the conflicting relationships identified previously in the literature and related clinical implications. We hypothesized that higher levels of shame would perpetuate substance use, and that continued substance use would, in turn, potentiate higher levels of shame. On the other hand, we hypothesized that guilt would not be bidirectionally related to the perpetuation of substance use, given the lack of associations found between guilt-proneness and substance use problems. Finally, we hypothesized that greater positive emotion would enable more adaptive responses to negative self-conscious emotions, resulting in reductions in substance use.

## Methods

This randomized controlled trial (RCT) enrolled and randomized 110 sexual minority men living with HIV who had biologically confirmed methamphetamine use in the greater San Francisco Bay Area. We have previously reported the efficacy of the positive emotion intervention delivered during contingency management for achieving durable and clinically meaningful reductions in HIV viral load [42]. All procedures were approved by the Institutional Review Board for the University of California, San Francisco with reliance agreements from the University of Miami and Northwestern University. All participants provided written informed consent prior to enrollment.

To be enrolled in the RCT, individuals had to be at least 18 years of age; self-identify as a sexual minority male; provide medical proof of seropositive status (i.e., letter of diagnosis, prescription of ART regimen other than Truvada); and provide a supervised urine or hair sample that was reactive for methamphetamine metabolites [43]. Participants completed a screening visit that included self-report measures as well as urine and hair toxicology screening to verify recent methamphetamine use. Approximately one week later, a baseline assessment was conducted

that included peripheral venous blood samples to measure HIV disease markers and urine samples to measure stimulant use (i.e. methamphetamine or cocaine) in the past 72 hours. Follow-up assessments were conducted at 3, 6, 12, and 15 months. Among the 110 participants randomized, follow-up rates at 3 (89%), 6 (88%), 12 (80%), and 15 (71%) months were acceptable with no significant differences in attrition between the experimental conditions.

## Measures

**Demographics and health status.** Participants self-reported their demographics including age, race, ethnicity, education, income, housing stability, sexual orientation. We consolidated race and ethnicity into: Black non-Latinx, White non-Latinx, Asian non-Latinx, other non-Latinx, and Latinx. Education was reported as less than high school, high school graduate, some college/trade school, college graduate, or graduate degree. Income was reported as <$4,999, $5,000 - $11,999, $12,000 - $15,999, $16,000 - $24,999, $25,000 - $34,999, $35,000 - $49,999, $50,000 - $74.999, $75,000 - $99,999, $100,000 - $124,999, and $125,000+. Time since HIV diagnosis was calculated based on self-reported diagnosis date.

**Substance use.** Participants reported the number of days they used specific substances in the past 30 days at all six time points. Stimulant use included powder cocaine, crack-cocaine, or methamphetamine. Where participants reported using multiple stimulants, the stimulant used the greatest number of days was selected for the composite measure of stimulant use in the past 30 days. Alcohol use included number of days participants reported drinking to intoxication in the past 30 days. Other substance use included heroin, methadone, painkillers, barbiturates, benzodiazepines, hallucinogens, inhalants, ketamine, MDMA, and GHB. Where participants reported using multiple other substances, the substance used with the greatest number of days was selected for the other substances in the past 30 days. Injection drug use was based on responses to the question, "On how many days in the past 30 days did you inject any substances?" Those who did not endorse recent injection drug use were assigned a "0."

**Negative self-conscious emotion.** We measured negative self-conscious emotion using the Differential Emotions Scale-IV [44, 45] at all six time points, which included three questions related to shame (*"In your daily life, how often do you feel like people laugh at you?; "In your daily life, how often do you feel embarrassed when anybody sees you a make a mistake?"*; and *"In your daily life, how often do you feel like people always look at you when anything goes wrong?")* and three related to guilt (*"In your daily life, how often do you feel regret?"; "In your daily life, how often do you feel like you did something wrong?"*; and *"In your daily life, how often do you feel like you ought to be blamed for something?")*. The response scale ranged from 1 [rarely or never] to 5 [very often] with higher average scores reflecting greater shame (Cronbach's alpha = 0.76) or guilt (Cronbach's alpha = 0.84).

**Positive emotion.** The Differential Emotions Scale, which was modified to include additional positive emotion items [43, 46] was measured at time 1. Participants rated how frequently they felt a particular emotion in the past week from 0 (never) to 4 (most of the time). The positive emotion measure included 14 items (Cronbach's alpha = 0.89).

**Depressive symptoms.** The Center for Epidemiologic Studies Depression 20-item Scale (CES-D; [47]) was administered to measure depressive symptoms at time 1. Participants rated how frequently they experienced symptoms of depression over the past week from 0 (rarely) to 3 (most or all of the time; Cronbach's alpha = 0.90).

## Data analysis

All preliminary analyses were conducted in SAS version 9.4 [48]. Descriptive statistics were obtained for all variables included in the analysis. Normality was examined for the variables of

interest (shame, guilt, stimulant use, alcohol use to intoxication, other drug use, and injection drug use) to determine type of distribution and estimation used. Given that all substance use variables were counts (number of days used in the past 30 days), maximum likelihood estimation with robust standard errors and Monte Carlo integration was used. Poisson, negative binomial, zero-inflated Poisson, and zero-inflated negative binomial models were estimated time period by time period and fit indices compared (AIC, SBC) to determine the best fitting distribution. The negative binomial distribution was determined to be the best fit across all substance use variables and used for all pathways predicting substance use.

To examine the relationships between shame and substance use and guilt and substance use, parallel process latent growth curve modeling was used in *Mplus* version 8.3 [49]. Using data from the six time points (i.e., screening, baseline, 3 months, 6 months, 12 months, and 15 months), unconditional latent growth curve models were specified for shame, guilt, and the substance use variables. Missing data were handled using full information maximum likelihood estimation (FIML); cases with both complete and incomplete data were included in the analysis via direct robust maximum likelihood estimation, thus all available observations were used in all models. Final models used $n = 109$ due to one individual missing all values. For each growth curve, a latent intercept (representing initial levels at the first time point) and a latent slope (representing change over time) were specified. Latent factors were fitted as random effects to allow for individual variation. Fit indices were examined for the continuous variables of shame and guilt (Chi-Square test of exact model fit ($p > .05$), Root Mean Squared Error of Approximation [RMSEA; $< .06$], comparative fit index [CFI; $> .95$]). These model fit indices are not available for the substance use count variables (negative binomial distribution).

The parallel process latent growth curve models were then fit to simultaneously examine cross-sectional and longitudinal bidirectional effects between shame and each substance use variable (see Fig 1A for graphical depiction of models). Time invariant covariates of age, being a person of color, income, and sexual orientation were associated with missingness at various timepoints, thus were included as covariates in the models. Additionally, although not associated with missingness, time since HIV diagnosis and unstable housing were also included as *a priori* covariates. We also chose to adjust for time 1 depressive symptoms due to depression being associated with some missingness at some time points and to examine the extent to

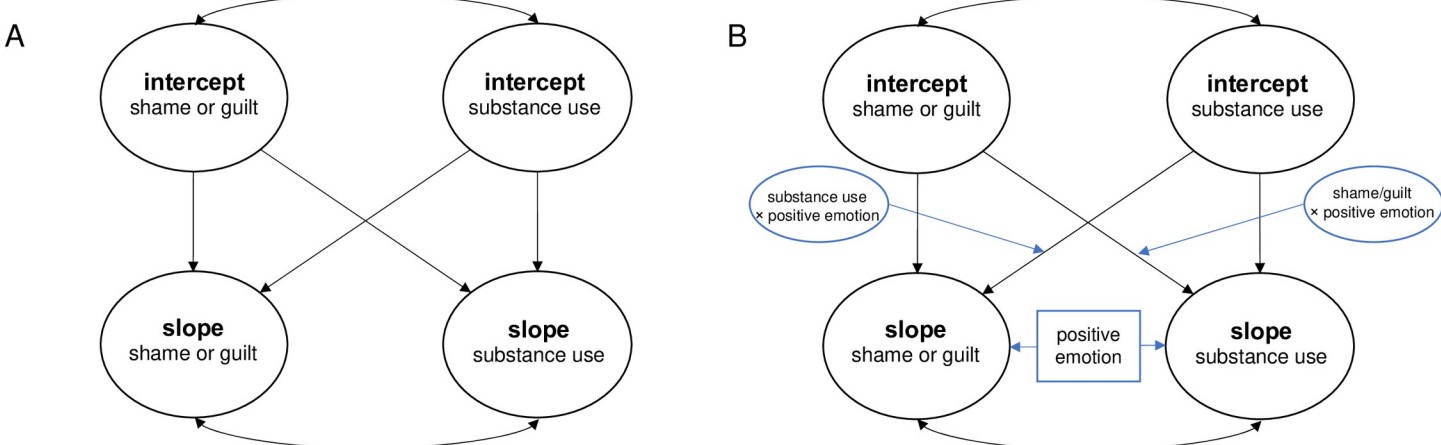

**Fig 1. Graphical depiction of models.** Notes. The first model (A) represents the initial models tested and the second model (B) represents the models with all the potential paths added for the examination of positive emotion as a moderator. Moderation was explored for any significant diagonal path in the first model (A). The main effects of positive emotion were first examined (including for the intercept to explore cross-sectional association) and then interaction terms entered (with main effect on intercept omitted). Intercepts and slopes were also regressed on covariates not shown in the model. Models also controlled for time invariant covariates including age, being a person of color, income, sexual orientation, time since HIV diagnosis, unstable housing, treatment condition, and time 1 depression.

which there are specific bidirectional associations between self-conscious emotions and indices of substance use over and above the well-established relationship with depressive symptoms (e.g., [2]). Given that the primary outcome analysis of the parent RCT found intervention-related changes in positive emotion (primary target of intervention) and stimulant use at 6 and 12 months [42], treatment group (randomized as part of the parent RCT) was also included as a covariate for the slopes.

The correlations between the intercepts represents the cross-sectional relationship between shame and substance use (i.e., are initial levels of shame correlated with initial levels of substance use?). Direct regression paths from the intercept of shame to the slope of substance use and the intercept of substance use to the slope of shame represent unidirectional effects (i.e., do initial level of shame predict the change of substance use over time and do initial levels of substance use predict the change of shame over time?). However, if both unidirectional direct paths were significant, this represents a bidirectional relationship. The correlation between the slopes also represents a bidirectional effect (i.e., do both variables move together over time?). Models controlled for the direct paths from the intercept of shame to the slope of shame and from the intercept of substance use to the slope of substance use (i.e., initial values influence trajectories within the same variable). Similar analyses were run replacing shame with guilt.

Positive emotion measured at time 1 (i.e., not confounded by treatment effects from parent RCT since this was a primary target) was then examined as a moderator for any significant diagonal directional paths in a model (see Fig 1B; i.e., does an individual's level of positive emotion change the relationships between shame or guilt and substance use). First, main effects of positive emotion were examined followed by entry of interaction terms. Interaction terms were defined using the XWITH command which creates a latent interaction term, which is needed for an interaction between a latent factor and an observed variable in a structural equation model [50]. Any significant interaction was then probed for simple slopes by rerunning the model with a centered moderator at +1 standard deviation (SD) above the mean and -1 SD below the mean. The same models were run replacing shame with guilt. Alpha was set to 0.05. For correlational paths, standard errors (*SE*) and *p* values are reported for the standardized beta coefficient (*β*). For the direct regression and moderation paths, *SE* and *p* values are reported for the unstandardized regression coefficient (*b*).

## Results

### Participant characteristics

Table 1 presents the study sample characteristics. Overall, participants had a mean age of 43 years (SD = 9, range 23 to 59), identified as a person of color (57.3%), identified as exclusively gay (76%), had more than a high school education (77%), had an income less than $16,000 (65%), were stably housed (91%), and had been diagnosed with HIV for 13 years (SD = 9, range = .04 to 36.15). Table 2 presents the study sample substance use characteristics. While eligibility criteria required that all participants had biologically verified methamphetamine use in the past 3–4 months, when asked about substance use in the past 30 days on self-report measures at baseline, 86% reported stimulant use, 26% reported alcohol use to intoxication, 71% reported other drug use, and 42% reported injection drug use. Notably, almost the entire sample (90%) were engaged in polysubstance use at baseline (i.e., actively using more than one drug type excluding alcohol to intoxication).

### Unconditional latent growth models

Results are presented in Table 3. Model fit indices revealed adequate model fit for shame (($\chi^2$ [16]) = 16.91, *p* = .391, RMSEA = 0.02, CFI = 0.99) and guilt (($\chi^2$ [16]) = 22.76, *p* = .121,

**Table 1. Participant characteristics N = 110.**

| | | *M or n* | *(SD or %)* |
|---|---|---|---|
| Age (years) | | 43.17 | (8.85) |
| Race/ethnicity | | | |
| | Black, non-Latinx | 18 | (16.4%) |
| | White, non-Latinx | 47 | (42.7%) |
| | Asian, non-Latinx | 3 | (2.7%) |
| | other, non-Latinx | 10 | (9.1%) |
| | Latinx | 32 | (29.1%) |
| Identifies as exclusively gay | | 83 | (75.5%) |
| Education | | | |
| | less than high school | 8 | (7.3%) |
| | high school graduate | 17 | (15.5%) |
| | some college/trade school | 57 | (51.8%) |
| | college graduate | 17 | (15.5%) |
| | graduate degree | 11 | (10.0%) |
| Income | | | |
| | < $4,999 | 16 | (14.5%) |
| | $5,000 - $11,999 | 28 | (25.5%) |
| | $12,000 - $15,999 | 27 | (24.5%) |
| | $16,000 - $24,999 | 12 | (10.9%) |
| | $25,000 - $34,999 | 9 | (8.2%) |
| | $35,000 - $49,999 | 9 | (8.2%) |
| | $50,000 - $69,000 | 7 | (6.4%) |
| | $75,000 - $99,000 | 0 | (0.0%) |
| | $100,000 - $124,000 | 0 | (0.0%) |
| | $125,000+ | 1 | (0.9%) |
| Unstable housing | | 9 | (9.1%) |
| Years since HIV diagnosis | | 12.94 | (8.61) |
| Positive Emotion[1] | | 32.3 | (8.2) |
| Depressive symptoms[2] | | 24.2 | (12.5) |
| Shame[3] | | | |
| | Screening | 2.56 | (0.93) |
| | Baseline | 2.61 | (0.90) |
| | 3-month | 2.43 | (0.79) |
| | 6-month | 2.30 | (0.79) |
| | 12-month | 2.34 | (0.86) |
| | 15-month | 2.26 | (0.83) |
| Guilt[3] | | | |
| | Screening | 2.82 | (1.01) |
| | Baseline | 2.73 | (0.97) |
| | 3-month | 2.58 | (0.87) |
| | 6-month | 2.44 | (0.95) |
| | 12-month | 2.49 | (0.95) |
| | 15-month | 2.39 | (0.76) |

Notes.

[1] Response scale ranging from 0 to 4 with higher summed scores (range 0–56) reflecting greater positive emotion

[2] Response scale ranging from 0 to 3 with summed scores (range 0–60) reflecting greater depression (scores = > 16 indicate clinically significant depression)

[3] Response scale ranging from 1 to 5 with higher average scores reflecting greater shame/guilt.

**Table 2. Participant substance use characteristics N = 110.**

| | | *M or n* | *(SD or %)* | Any use past 30 days | |
|---|---|---|---|---|---|
| | | | | *n* | *(%)* |
| Ever alcohol to intoxication during study | | 76 | (69.1%) | | |
| Past 30-day alcohol intoxication days | | | | | |
| | Screening | 1.79 | (5.15) | 29 | (26.4%) |
| | Baseline | 1.64 | (4.87) | 27 | (24.6%) |
| | 3-month | 1.56 | (4.56) | 21 | (21.4%) |
| | 6-month | 1.44 | (3.60) | 26 | (27.1%) |
| | 12-month | 0.99 | (2.67) | 16 | (18.2%) |
| | 15-month | 1.43 | (4.41) | 20 | (26.0%) |
| Ever stimulant use during study | | 108 | (98.2%) | | |
| Past 30-day stimulant use[1] days | | | | | |
| | Screening | 9.75 | (9.55) | 95 | (86.4%) |
| | Baseline | 8.37 | (9.27) | 89 | (80.9%) |
| | 3-month | 6.87 | (8.62) | 71 | (72.5%) |
| | 6-month | 6.38 | (8.49) | 68 | (70.8%) |
| | 12-month | 7.39 | (9.61) | 62 | (70.5%) |
| | 15-month | 7.12 | (9.50) | 49 | (63.6%) |
| Ever other drug use during study | | 101 | (91.8%) | | |
| Past 30-day other drug use[2] days | | | | | |
| | Screening | 6.88 | (9.67) | 78 | (70.9%) |
| | Baseline | 4.31 | (7.81) | 59 | (53.6%) |
| | 3-month | 4.67 | (8.34) | 52 | (53.1%) |
| | 6-month | 5.01 | (8.65) | 48 | (50.0%) |
| | 12-month | 3.93 | (7.10) | 42 | (47.7%) |
| | 15-month | 3.83 | (7.57) | 37 | (48.1%) |
| Ever IDU during study | | 79 | (71.8%) | | |
| Past 30-day IDU days | | | | | |
| | Screening | 4.57 | (8.22) | 46 | (41.8%) |
| | Baseline | 3.97 | (7.38) | 44 | (40.0%) |
| | 3-month | 1.73 | (4.30) | 32 | (32.7%) |
| | 6-month | 2.03 | (4.84) | 32 | (33.3%) |
| | 12-month | 3.03 | (6.55) | 30 | (34.1%) |
| | 15-month | 2.49 | (6.42) | 22 | (28.6%) |
| Past 30-day polysubstance use[3] at baseline | | 99 | (90.0%) | | |

[1]Cocaine, crack, methamphetamine, or other stimulants

[2]heroin, methadone, pain killers, barbiturates, benzodiazepines, hallucinogens, inhalants, ketamine, MDMA, or GHB

[3]used 2+ substances including marijuana and excluding alcohol; IDU = injection drug use.

RMSEA = 0.06, CFI = 0.97). Results indicated that shame ($b$ = -0.06, $SE$ = 0.02, $p$ = .004), guilt ($b$ = -0.07, $SE$ = 0.02, $p < .0001$), and stimulant use ($b$ = -0.17, $SE$ = 0.06, $p$ = .005) significantly decreased over time. Alcohol use to intoxication ($b$ = 0.06, $SE$ = 0.36, $p$ = .720), other drug use ($b$ = -0.15, $SE$ = 0.08, p = .062), and injection drug use ($b$ = -0.28, $SE$ = 0.15, p = .054) remained stable over time.

**Table 3. Unconditional latent growth curve models, N = 110.**

| | | β | b | SE(b) | p |
|---|---|---:|---:|---:|---:|
| **Shame** | | | | | |
| Fixed effects | Intercept | 3.41 | 2.58 | 0.08 | < .0001 |
| | Slope | -0.37 | -0.06 | 0.02 | .004 |
| Random effects | Intercept | 1.00 | 0.57 | 0.10 | < .0001 |
| | Slope | 1.00 | 0.02 | 0.01 | < .0001 |
| **Guilt** | | | | | |
| Fixed effects | Intercept | 3.47 | 2.78 | 0.09 | < .0001 |
| | Slope | -0.58 | -0.07 | 0.02 | < .0001 |
| Random effects | Intercept | 1.00 | 0.64 | 0.12 | < .0001 |
| | Slope | 1.00 | 0.01 | 0.01 | .017 |
| **Stimulant use** | | | | | |
| Fixed effects | Intercept | 1.88 | 1.89 | 0.14 | < .0001 |
| | Slope | -0.49 | -0.17 | 0.06 | .005 |
| Random effects | Intercept | 1.00 | 1.01 | 0.30 | .001 |
| | Slope | 1.00 | 0.13 | 0.05 | .006 |
| **Alcohol to intoxication** | | | | | |
| Fixed effects | Intercept | -0.80 | -2.43 | 0.36 | .118 |
| | Slope | 0.15 | 0.06 | 0.16 | .720 |
| Random effects | Intercept | 1.00 | 9.33 | 2.16 | < .0001 |
| | Slope | 1.00 | 0.14 | 0.08 | .078 |
| **Other drug use** | | | | | |
| Fixed effects | Intercept | 0.45 | 0.81 | 0.24 | .001 |
| | Slope | -0.37 | -0.15 | 0.08 | .062 |
| Random effects | Intercept | 1.00 | 3.22 | 0.60 | < .0001 |
| | Slope | 1.00 | 0.15 | 0.05 | .005 |
| **Injection drug use** | | | | | |
| Fixed effects | Intercept | -0.25 | -0.76 | 0.49 | .119 |
| | Slope | -0.46 | -0.28 | 0.15 | .057 |
| Random effects | Intercept | 1.00 | 9.53 | 1.96 | < .0001 |
| | Slope | 1.00 | 0.38 | 0.15 | .010 |

*Notes.* Model fit indices revealed adequate model fit for shame ($\chi^2$ [16]) = 16.91, $p$ = .391, RMSEA = 0.02, CFI = 0.99) and guilt ($\chi^2$ [16]) = 22.76, $p$ = .121, RMSEA = 0.06, CFI = 0.97); absolute fit indices are not available for the substance use count variables (negative binomial distribution)

## Parallel process growth models

Results are presented in Table 4. Of note, the positive emotion intervention tested in the original study did result in significant reductions in self-reported stimulant use and increases in positive emotion [42]. Negative self-conscious emotions (shame and guilt) were not assessed in the original analyses.

**Stimulant use.** Shame had a longitudinal unidirectional relationship with stimulant use and guilt had a longitudinal bidirectional relationship with stimulant use. The intercept of shame significantly predicted the slope of stimulant use ($\beta$ = 0.48, $b$ = 0.23, $SE$ = 0.11, $p$ = .041), but the intercept of stimulant use did not predict the slope of shame. In other words, those who had higher initial levels of shame had slower decreases in stimulant use over time, but reverse directionality was not significant such that initial levels of stimulant use did not influence the trajectory of shame. Additionally, there was a significant positive relationship between the slopes of guilt and stimulant use ($b$ = 0.03, $\beta$ = 0.85, $SE$ = 0.18, $p$ < .0001) indicating they

**Table 4. Parallel process latent growth curve models, N = 109.**

| | Stimulant use | | | | Alcohol use to intoxication | | | | Other drug use | | | | Injection drug use | | | |
|---|---|---|---|---|---|---|---|---|---|---|---|---|---|---|---|---|
| | *β* | *b* | *SE* | *p* | *β* | *b* | *SE* | *p* | *β* | *b* | *SE* | *p* | *β* | *b* | *SE* | *p* |
| **SHAME** | | | | | | | | | | | | | | | | |
| **Cross-sectional association** | | | | | | | | | | | | | | | | |
| Intercept shame ↔ Intercept substance use | 0.01 | 0.01 | 0.18 | .937 | 0.13 | 0.19 | 0.15 | .396 | -0.05 | -0.05 | 0.15 | .705 | -0.17 | -0.26 | 0.14 | .222 |
| **Longitudinal bidirectional effects** | | | | | | | | | | | | | | | | |
| Slope shame ↔ Slope substance use | 0.06 | 0.002 | 0.21 | .774 | 0.96 | 0.01 | 1.67 | .565 | -0.07 | -0.002 | 0.25 | .783 | 0.11 | 0.01 | 0.25 | .678 |
| Intercept of shame → Slope substance use | 0.48 | 0.23 | 0.11 | **.041** | -0.71 | -0.36 | 0.19 | .058 | 0.43 | 0.22 | 0.11 | .056 | 0.49 | 0.39 | 0.21 | .061 |
| Intercept of substance use → Slope shame | 0.06 | 0.01 | 0.02 | .723 | 0.10 | 0.01 | 0.01 | .476 | 0.29 | 0.02 | 0.01 | **.041** | 0.12 | 0.01 | 0.01 | .357 |
| **GUILT** | | | | | | | | | | | | | | | | |
| **Cross-sectional association** | | | | | | | | | | | | | | | | |
| Intercept guilt ↔ Intercept substance use | -0.08 | -0.04 | 0.16 | .600 | 0.29 | 0.47 | 0.14 | **.040** | 0.02 | 0.02 | 0.14 | .883 | 0.04 | 0.06 | 0.15 | .805 |
| **Longitudinal bidirectional effects** | | | | | | | | | | | | | | | | |
| Slope guilt ↔ Slope substance use | 0.85 | 0.03 | 0.18 | **< .0001** | -0.99 | -0.51 | 0.03 | **< .0001** | 0.70 | 0.01 | 0.61 | .248 | 0.99 | 0.03 | 0.57 | .084 |
| Intercept of guilt → Slope substance use | -0.10 | -0.04 | 0.07 | .575 | 0.19 | 0.19 | 0.18 | .283 | 0.02 | 0.01 | 0.10 | .945 | -0.04 | -0.03 | 0.17 | .841 |
| Intercept of substance use → Slope guilt | -0.12 | -0.02 | 0.02 | .304 | 0.01 | 0.001 | 0.01 | .813 | 0.24 | 0.02 | 0.01 | .156 | 0.03 | 0.001 | 0.01 | .866 |

Notes. Alpha < .05; double headed arrow = correlation (*SE* and *p* reported for standardized beta coefficients); single headed arrow = direct regression path (*SE* and *p* reported for unstandardized regression coefficients); models controlled for age, being a person of color, income, unstable housing, sexual orientation, time since HIV diagnosis, depressive symptoms, and the direct path between the intercept and slope within the same variable (e.g., intercept shame → slope shame); moderation results not shown

move together in time (i.e., as one decreases/increases, the other also decreases/increases, respectively). There were no cross-sectional associations between shame, guilt, and stimulant use.

Given the significant path from the intercept of shame to the slope of stimulant use, positive emotion was explored as a moderator of this relationship. There was a significant cross-sectional relationship such that higher positive emotion was associated with lower initial levels of stimulant use ($β$ = -0.46, $b$ = -0.06, $SE$ = 0.02, $p$ < .0001). There was not a main effect of positive emotion on the trajectory of stimulant use nor any interaction effects (i.e., positive emotion did not influence the relationship between shame and stimulant use).

**Alcohol use to intoxication.** Shame and alcohol use were not significantly associated cross-sectionally or longitudinally. However, guilt had a cross-sectional and longitudinal bidirectional relationship with alcohol use to intoxication. There was a significant positive relationship between the intercepts of guilt and alcohol use ($b$ = 0.47, $β$ = 0.29, $SE$ = 0.14, $p$ < .0001) indicating that as initial levels of guilt increased or decreased, initial levels of alcohol use also increased or decreased (and vice versa). Additionally, there was a significant negative relationship between the slopes of guilt and alcohol use ($b$ = -0.51, $β$ = -0.99, $SE$ = 0.03, $p$ < .0001) indicating that they move together in time, but in opposite directions (i.e., as one decreases, the other increases and vice versa).

**Other drug use and injection drug use.** Shame had a longitudinal unidirectional relationship with other drug use and guilt did not have any relationship with other drug use. The intercept of other substance use significantly predicted the slope of shame ($β$ = 0.29, $b$ = 0.02, $SE$ = 0.01, $p$ = .041), but the intercept of shame did not predict the slope of other substance use. In other words, those who had higher initial levels of other drug use had slower decreases in shame over time, but reverse directionality was not significant, such that initial levels of shame did not influence the trajectory of other drug use. There were no cross-sectional associations between shame, guilt, and other drug use.

Given the significant path from the intercept of shame to the slope of other drug use, positive emotion was explored as a moderator of this relationship. There were no main effects nor any interaction effects (i.e., positive emotion did not influence the relationship between other drug use and shame). Shame and guilt did not have any cross-sectional or longitudinal associations with injection drug use.

## Discussion

These innovative results are the first we are aware of to identify bidirectional relationships between negative self-conscious emotions and substance use, including differential relationships by type of substance use. While we did not find evidence to support all of our hypotheses, our findings indicated that high levels of shame may delay the pace of stimulant use reduction and that as guilt decreases or increases stimulant use correspondingly decreases or increases, respectively. Notably, these relationships were not attributable to depressive symptoms. While these findings are consistent with existing literature, indicating the perpetuating relationships between shame and guilt and substance use (e.g., [7, 19, 20]), they add to the existing literature by demonstrating that shame and guilt may be critical barriers for stimulant use recovery, beyond the previously identified relationships between depression and stimulant use. Further, while more work is needed, these nuanced results may be consistent with meta-analytic finding indicating that the likelihood of shame eliciting an adaptive rather than a maladaptive response is dependent on individuals' perception of their ability to repair a positive sense of self [10]. Specifically, those with higher levels of shame may not have the capacity to repair their positive sense of self, and therefore, may experience a slower reduction in stimulant use. Initial levels of stimulant use were not associated with decreases in shame over time, indicating that stimulant use may not be initiating the cyclical relationship with shame, but rather the initial levels of shame may be pre-existing or attributable to another cause. However, guilt appears to coincide with both increases and decreases in stimulant use across participants, potentially consistent with the conflicting literature related to guilt and substance use [17, 18]. We also did not identify any cross-sectional relationships between stimulant use and shame or guilt, emphasizing the importance of longitudinal analyses of these complex relationships [4]. Although, positive emotion was not associated with trajectories of stimulant use, shame, or guilt, higher level of positive emotion was associated with lower initial levels of stimulant use, consistent with theorized associations of positive emotion and adaptive behavior (e.g., [33]).

In addition to the results related to stimulant use, our results indicate differential relationships between other types of substance use and shame and guilt both consistent and inconsistent with the literature. In relation to alcohol use to intoxication, shame was neither cross-sectionally nor longitudinally associated with alcohol use. As most research investigating shame and substance use focuses on alcohol [4], it is notable that shame was not associated with alcohol use to intoxication in this sample. However, guilt was associated with alcohol use to intoxication. Cross-sectionally, higher initial levels of guilt were associated with higher initial levels of alcohol use to intoxication, and lower levels of guilt were associated with lower levels of intoxication, consistent with the literature indicating relationships between guilt-proneness and alcohol use disorders [18]. Longitudinally, as guilt increased, alcohol use to intoxication decreased and vice versa. This finding may be indicative of the conflicting findings related to guilt, including adaptive or protective aspects of guilt, consistent with evidence indicating that guilt-proneness is associated with harm-avoidant behaviors such as limiting drinking [16] as well as being associated with alcohol dependence [18].

In relation to other drug use, those reporting higher levels of other drug use initially had slower decreases in shame, but shame did not predict the trajectory of other substance use.

This may indicate that, unlike stimulant use, other drug use may be initiating and perpetuating shame. As stimulant use may be more closely associated with other stigmatized identities (e.g., sexual orientation or HIV-status) in this sample of sexual minority men living with HIV, shame may initiate that cycle, whereas other substance use may be the source of shame for some, thereby initiating the perpetuation of shame. Neither guilt nor positive emotion were associated with other drug use in this sample. Further, there were no cross-sectional or longitudinal relationships between shame or guilt and injection drug use in this sample.

It is important to interpret these findings in the context of the sample of sexual minority men living with HIV who use methamphetamine. In addition to substance use-related stigma, and related shame and guilt, participants in this study may also be affected by stigmas associated with their intersecting identities, including being sexual minorities and living with HIV. These stigmas may impact shame or guilt-proneness. For example, evidence indicates that internalized homonegativity is positively associated with shame-proneness [51]. Further, substance use, particularly stimulant use, may be associated with sexual experiences in the context of substance use (e.g., chem-sex), which likely elicits both positive and negative emotions, including negative self-conscious emotions, and may impact substance use behaviors. For example, in a study of 83 Filipino Americans (78% MSM) using methamphetamines, greater shame-related substance use predicted lower frequency of methamphetamine use in the preceding 30 days, but not frequency of methamphetamine use before sex [52]. Living with intersecting stigmatized identities may account for the differences in relationships identified between substances. As stimulant use is frequently associated with chemsex, the identified association between initial level of shame and slower reductions in stimulant use may indicate that shame, potentially attributable to other factors (e.g., sexual orientation or HIV status in the context of sex), initiates and perpetuates stimulant use. In contrast, the relationship identified between shame and other substance use indicates that initial levels of shame were not associated with other drug use, but initial levels of drug use were associated with shame over time. This finding may indicate that other drug use may be the impetus for shame. While these results have implications across individuals who use substances, additional research is needed to better understand the intersectional relationships between negative self-conscious emotions, positive emotions, and substance use among sexual minority men living with HIV. There is also a clear need for clinical research examining the bidirectional associations between self-conscious emotions and substance use in the broader population of people living with stimulant use disorders.

While these are the first results we are aware of that identify bidirectional relationships between negative self-conscious emotions and substance use among sexual minority men living with HIV, there are several limitations. First, the sample is relatively modest, which may have influenced our results (e.g., the differential findings across substances). Samples between 100–150 are frequently considered the minimum for conducting structural equation models (e.g., [53]); while our sample meets this minimum recommendation, we acknowledge its relatively modest size. Although, recent work suggests that when using robust maximum likelihood estimation for latent growth curves, as the current study did, even small samples of N<100 provide accurate estimates even in the context of non-normality and missing data [54]. Second, the sample is taken from a RCT designed to increase positive emotion, thereby limiting the generalizability beyond individuals willing to engage in this type of intervention. Third, participants' ability to differentiate shame from guilt is unknown; it is possible that shame and guilt were conflated by participants. This limits our ability to draw conclusions about the distinctions between the two negative self-conscious emotions in relation to substance use and positive emotion. Further research is needed to ensure accurate measurement of specific negative self-conscious emotions. Finally, the intervention's effect on stimulant use and positive emotion may have affected the identified associations.

In conclusion, our results offer novel insights into the complex and nuanced relationships between substance use and negative self-conscious emotions, including differential relationships by type of substance use in a sample of sexual minority men living with HIV who use methamphetamine. Most notably, our findings indicate that high levels of shame may delay the pace of stimulant use reduction and guilt may sustain stimulant use, identifying these emotions as prominent barriers to stimulant use recovery. Additionally, we identified differential relationships between shame and guilt and other substance use. Specifically, higher levels of guilt, but not shame, were associated with more frequent alcohol use to intoxication cross-sectionally but less longitudinally, indicating that guilt may be positively associated with recent alcohol use to intoxication but may be protective over time. Finally, those reporting higher levels of other drug use initially had slower decreases in shame, but shame did not predict the trajectory of other substance use. While additional work is needed to enhance our understanding of the nuanced relationships between substance use and negative self-conscious emotions to more effectively intervene and ultimately reduce substance use, these results provide compelling novel insights into the complex relationships between substance use and behaviorally influential negative self-conscious emotions.

## Supporting information

**S1 File.**
(ZIP)

## Author Contributions

**Conceptualization:** Abigail W. Batchelder, Tiffany R. Glynn, Judith T. Moskowitz, Adam W. Carrico.

**Data curation:** Samantha Dilworth.

**Formal analysis:** Tiffany R. Glynn.

**Funding acquisition:** Adam W. Carrico.

**Investigation:** Judith T. Moskowitz, Adam W. Carrico.

**Methodology:** Tiffany R. Glynn, Torsten B. Neilands, Samantha Dilworth, Adam W. Carrico.

**Resources:** Adam W. Carrico.

**Software:** Sara L. Rodriguez.

**Supervision:** Adam W. Carrico.

**Validation:** Torsten B. Neilands.

**Writing – original draft:** Abigail W. Batchelder, Tiffany R. Glynn, Sara L. Rodriguez.

**Writing – review & editing:** Abigail W. Batchelder, Tiffany R. Glynn, Judith T. Moskowitz, Torsten B. Neilands, Samantha Dilworth, Sara L. Rodriguez, Adam W. Carrico.

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
