## [Decision Letter · Decision Letter 0]

13 Dec 2021

PONE-D-21-12470The shame spiral of addiction: Negative self-conscious emotion and substance usePLOS ONE

Dear Dr. Batchelder,

Thank you for submitting your manuscript to PLOS ONE. After careful consideration, we feel that it has merit but does not fully meet PLOS ONE’s publication criteria as it currently stands. Therefore, we invite you to submit a revised version of the manuscript that addresses the points raised during the review process.

In addition to addressing the reviewer comments below, please provide more information on the pattern of missingness in the data, especially with respect to attrition. Any variables predictive of missingness should be included as auxiliary variables in the latent growth models to correct for any potential bias.

We look forward to receiving your revised manuscript.

Kind regards,

Matthew J. Gullo

Academic Editor

PLOS ONE

Additional Editor Comments (if provided):

More information on the pattern of missingness in the data is required, especially with respect to attrition. Any variables predictive of missingness should be included as auxiliary variables in the latent growth models to correct for any potential bias.

Journal Requirements:

Reviewers' comments:

Reviewer's Responses to Questions

**Comments to the Author**

1. Is the manuscript technically sound, and do the data support the conclusions?

Reviewer #1: Partly

Reviewer #2: Yes

2. Has the statistical analysis been performed appropriately and rigorously? 

Reviewer #1: Yes

Reviewer #2: I Don't Know

3. Have the authors made all data underlying the findings in their manuscript fully available?

Reviewer #1: Yes

Reviewer #2: No

4. Is the manuscript presented in an intelligible fashion and written in standard English?

Reviewer #1: Yes

Reviewer #2: Yes

5. Review Comments to the Author

Reviewer #1: Using parallel process latent growth curve modeling, the investigators assessed bidirectional associations between shame and guilt and substance use (i.e., number of days in the past 30 used stimulants, alcohol to intoxication, other substances, or injected drugs) as well as the moderating role of positive emotion. The sample included 110 sexual minority cisgender men with biologically confirmed recent methamphetamine use, enrolled in a randomized controlled trial in San Francisco, CA.

The paper is well written and the models were well explained. Also the results appear to support the conclusions, most notably, the findings indicate that high levels of shame may delay the pace of stimulant use reduction and guilt may sustain stimulant use, identifying these emotions as prominent barriers to stimulant use recovery.

Often with the use of latent models one wonders about the adequacy of the sample of 110 with the number of paths established. The authors should comment on the statistical adequacy of the sample size in this context.

Reviewer #2: This study addresses the important topic of the bidirectional relationships among shame, guilt, and substance use. The introduction provides a comprehensive literature review of the topic and clearly sets up the gaps in the literature that this study will address. However, this section would be strengthened by stronger motivation of the clinical significance of this question. How will delineating the relationships among shame, guilt, and substance use inform interventions and improve patient care?

Were substance use, emotions and depressive symptoms measured at each time point (baseline, 3, 6, 12, 15 months)? It seems the answer is yes from the description of the analysis, but this should be made clear in the measures section.

The point about the follow-up rates in the 110 participants randomized is repeated at two points in the manuscript; one can be removed.

The latent growth curve models used individuals randomized to the intervention or control group in the parent study. Given that the intervention group received a positive emotion intervention, could this introduce bias into the present analysis, even with the covariate for the slopes? It is not entirely clear to me what this covariate accomplishes.

Page 14, “Of note, in the original analysis, there were no treatment effects found for stimulant use…” is unclear; does this mean that the positive emotion intervention tested in the original study had no effects on stimulant use (but did increase parallel emotion?)? Clarify so that readers aren’t forced to go to the study cited to understand this point.

How do the authors interpret the differential relationships by type of substance use – why would these relationships differ across substances? Are these meaningful, or related to low statistical power?

6. PLOS authors have the option to publish the peer review history of their article (what does this mean?). If published, this will include your full peer review and any attached files.

Reviewer #1: No

Reviewer #2: No

---

## [Author Response · Author response to Decision Letter 0]

19 Feb 2022

Dear Drs. Matthew Gullo and Emily Chenette,

Thank you for considering our revised enclosed research article, “The Shame Spiral of Addiction: Negative Self-Conscious Emotion and Substance Use” for publication in PLOS ONE. We appreciate the reviewers’ comments and suggestions and have addressed each below. We believe this manuscript is stronger because of these edits.

Editor’s Comment:

1. More information on the pattern of missingness in the data is required, especially with respect to attrition. Any variables predictive of missingness should be included as auxiliary variables in the latent growth models to correct for any potential bias.

-In addition to the current sentences indicating attrition on page 9 (“Among the 110 participants randomized, follow-up rates at 3 (89%) 6 (88%), 12 (80%), and 15 (71%) months were acceptable with no significant differences in attrition between the experimental conditions.”) we expanded the sentence on page 11 to explain our approach to missing data more thoroughly, “Missing data were handled using full information maximum likelihood estimation (FIML); cases with both complete and incomplete data were included in the analysis via direct robust maximum likelihood estimation, thus all available observations were used in all models.”). Further, we clarified that part of our a prior covariate choice included testing to see if they were predictive of missingness which supported including them in the models (see bottom of page 11 and top of page 12 for text). We have also added text to our Figure note to clarify that models controlled for these auxiliary variables. 

Reviewer 1:

2. The paper is well written and the models were well explained. Also the results appear to support the conclusions, most notably, the findings indicate that high levels of shame may delay the pace of stimulant use reduction and guilt may sustain stimulant use, identifying these emotions as prominent barriers to stimulant use recovery.

- We thank this reviewer for these comments.

3. Often with the use of latent models one wonders about the adequacy of the sample of 110 with the number of paths established. The authors should comment on the statistical adequacy of the sample size in this context.

- We appreciate this comment and have expanded our comment in the discussion about the adequacy of the sample size and the related limitations of using this method. On page 19, we added: “Samples between 100-150 are frequently considered the minimum for conducting structural equation models (e.g., Tabachnick & Fidell, 2001); while our sample meets this minimum recommendation, we acknowledge its relatively modest size. Although, recent work suggests that when using robust maximum likelihood estimation for latent growth curves, as the current study did, even small samples of N<100 provide accurate estimates even in the context of non-normality and missing data (Shi et al., 2021). “

Reviewer 2:

4. This study addresses the important topic of the bidirectional relationships among shame, guilt, and substance use. The introduction provides a comprehensive literature review of the topic and clearly sets up the gaps in the literature that this study will address. However, this section would be strengthened by stronger motivation of the clinical significance of this question. How will delineating the relationships among shame, guilt, and substance use inform interventions and improve patient care?

 - We have added a stronger rationale for the clinical significance of delineating the relationships between shame and guilt in relation to substance use to the introduction. On page 7-8 we have added: “We examine shame and guilt separately given the literature indicating their unique psychological experiences (5) and behavioral sequelae (6,10,11) as well as their differential associations with addiction-related behaviors (16, 17,18)… Investigating the causes, consequences, and moderators of shame and guilt separately in relation to substance use represents a critical step in clarifying the conflicting relationships identified previously in the literature and related clinical implications.”

5. Were substance use, emotions and depressive symptoms measured at each time point (baseline, 3, 6, 12, 15 months)? It seems the answer is yes from the description of the analysis, but this should be made clear in the measures section. 

 - We have clarified that substance use, shame, and guilt were measured at all time points and that positive emotion and depressive symptoms were measured at time 1 in the measures section. 

6. The point about the follow-up rates in the 110 participants randomized is repeated at two points in the manuscript; one can be removed.

- We have removed the second sentence related to follow up rates on page 11.

7. The latent growth curve models used individuals randomized to the intervention or control group in the parent study. Given that the intervention group received a positive emotion intervention, could this introduce bias into the present analysis, even with the covariate for the slopes? It is not entirely clear to me what this covariate accomplishes.

- To minimize the potential impact of the positive emotion intervention on the results, we only included positive emotion at the initial timepoint (prior to randomization) as a moderator in our models. Further, we included the treatment group as a covariate for the slopes to account for the variance in change over time related to being in the intervention. We believe this is an important covariate to include given that these data were collected as part of a parent RCT. We have further emphasized this in the limitations section on page 19. 

8. Page 14, “Of note, in the original analysis, there were no treatment effects found for stimulant use…” is unclear; does this mean that the positive emotion intervention tested in the original study had no effects on stimulant use (but did increase parallel emotion?)? Clarify so that readers aren’t forced to go to the study cited to understand this point.

 -We have clarified this point on page 15.

9. How do the authors interpret the differential relationships by type of substance use – why would these relationships differ across substances? Are these meaningful, or related to low statistical power?

 -We appreciate this inquiry and have added details to the discussion related to our cautious interpretation of these differences in the context of the existing literature. On page 17 we have explicitly indicated that our results were both “consistent and inconsistent with the existing literature” and have added the following to the limitations section on page 19, “First, the sample is relatively modest, which may have influenced our results (e.g., the differential findings across substances). Samples between 100-150 are frequently considered the minimum for conducting structural equation models (e.g., Tabachnick & Fidell, 2001); while our sample meets this minimum recommendation, we acknowledge its relatively modest size.”

Journal Comments:

- We have reviewed the PLOS ONE style requirements, including those for file naming and have confirmed that this submission is consistent with the requirements.

11. We note that the grant information you provided in the ‘Funding Information’ and ‘Financial Disclosure’ sections do not match. 

 -We have corrected these inconsistencies.

 - We have confirmed this.

13. In your Data Availability statement, you have not specified where the minimal data set underlying the results described in your manuscript can be found. 

-We have specified in greater detail where the minimal data set underlying the results can be found.

Thank you for considering this manuscript for PLOS ONE. 

Sincerely,

Abigail Batchelder, Ph.D., M.P.H. 

Behavioral Medicine, Department of Psychiatry 

Massachusetts General Hospital 

Harvard Medical School 

One Bowdoin Square, 7th Floor 

Boston, Massachusetts 

Phone: (917) 940-1283 

Fax: (617) 724-3726

Email: ABatchelder@mgh.harvard.edu

Alternative email: abby.batchelder@gmail.com

---

## [Editor Report · Decision Letter 1]

3 Mar 2022

The shame spiral of addiction: Negative self-conscious emotion and substance use

PONE-D-21-12470R1

Dear Dr. Batchelder,

We’re pleased to inform you that your manuscript has been judged scientifically suitable for publication and will be formally accepted for publication once it meets all outstanding technical requirements.

Kind regards,

Matthew J. Gullo

Academic Editor

PLOS ONE
---

## [Editor Report · Acceptance letter]

10 Mar 2022

PONE-D-21-12470R1 

The shame spiral of addiction: Negative self-conscious emotion and substance use 

Dear Dr. Batchelder:

I'm pleased to inform you that your manuscript has been deemed suitable for publication in PLOS ONE. Congratulations! Your manuscript is now with our production department. 

Kind regards, 

on behalf of

Assoc. Prof. Matthew J. Gullo 

Academic Editor

PLOS ONE